# ProxyFL: A Proxy-Guided Framework for Federated Semi-Supervised Learning

## Abstract

Federated Semi-Supervised Learning (FSSL) aims to collaboratively train a global model by leveraging unlabeled data and limited labeled data across clients in a privacy-preserving manner. In FSSL, data heterogeneity is a challenging issue, which exists both across clients (external heterogeneity) and within clients (internal heterogeneity). Most FSSL methods typically design fixed or dynamic weight aggregation strategies on the server (for external) or filter out low-confidence unlabeled samples directly by an empirical threshold to reduce mistakes in local client (for internal). But, the former is hard to precisely fit the real global category distribution due to external heterogeneity, and the latter results in fewer training participation of available samples in FL. To address these issues, we propose a proxy-guided framework called ProxyFL that focuses on simultaneously mitigating external and internal heterogeneity via a unified proxy. *I.e.*, we consider the learnable weights of classifier as proxy to simulate the category distribution both locally and globally. For external, we explicitly optimize global proxy to better fit the category distribution across clients; for internal, we include the discarded samples together with other samples into training based upon a positive-negative proxy pool without compromising wrong pseudo-labels. Insight experiments & theoretical analysis show that ProxyFL significantly boost the FSSL performance and convergence.

## 1 Introduction

The rapid advancement of edge devices and the Internet of Things (IoT) has led to a pressing need for decentralized training paradigms (Hoofnagle et al. (2019); Lim et al. (2020)). Federated learning (FL), a distributed machine-learning paradigm, facilitates multi-device collaborative learning without compromising data privacy, which shares only model updates rather than raw data (McMahan et al. (2017)). Most existing FL works assume that local data in clients are fully labeled, but this assumption does not hold in practical scenarios when data annotation is laborious, time-consuming, or expensive. To remedy these issues, Federated Semi-Supervised Learning (FSSL) has emerged, enabling clients to train models leveraging both limited labeled data and a large amount of unlabeled data, thereby improving the performance of global model. In FSSL, data heterogeneity exists both across clients (external heterogeneity) and within clients (internal heterogeneity). The former refers to the distribution discrepancy across different clients, while the latter arises from the local mismatch due to (1) imbalanced samples sizes across different categories and (2) distribution imbalance between labeled & unlabeled data.

Existing FSSL works primarily rely on consistency regularization between model prediction and pseudo-labels: To handle internal heterogeneity, most methods such as FedLabel (Cho et al. (2023)), FedDB (Zhu et al. (2024)) and SAGE (Liu et al. (2025)) typically filter high-confidence unlabeled samples for training, while excluding low-confidence ones to avoid introducing bias. FedDure (Bai et al. (2024)) leverages low-confidence samples by dynamically assigning them a smaller weight. However, the first three methods (FedLabel, FedDB and SAGE) lead to fewer data participation due to discarding low-confidence unlabeled data directly, while FedDure compromises incorrect pseudo-labels by setting smaller weights; For external heterogeneity, as the server has no access to local samples for data privacy, the aggregation weights are often calculated based on local dataset sizes (Bai et al. (2024); Liu et al. (2025)) or some implicit statistics of local samples (Cho et al. (2023); Zhu et al. (2024)), which may deviate from the global distribution across clients. These mo-

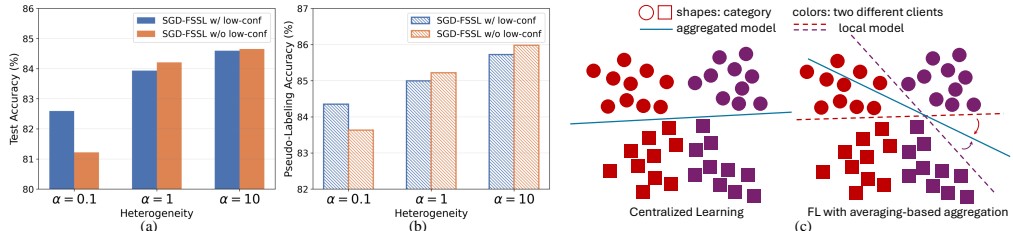

Figure 1: (a-b) Differences of test accuracy and pseudo-labeling accuracy under varying levels of heterogeneity (smaller $\alpha$ indicates greater heterogeneity). During each communication round, all clients are trained based on FedSGD (McMahan et al. (2017)) for one local epoch *w/* and *w/o* low-confidence samples, respectively. (c) Illustration of centralized learning and averaging-based decentralized FL approaches.

tivate us to pose the following questions: ① *Can the local model leverage low-confidence unlabeled samples without compromising the wrong pseudo-labels?* ② *Is it possible to explicitly fit the global distribution across clients in a privacy-preserving manner?*

To this end, we first conduct ablation experiments to study Question ①. Fig. 1(a-b) show test accuracy and pseudo-labeling accuracy under different levels of heterogeneity. We observe that simply discarding low-confidence samples and directly incorporating them with pseudo-labels exhibit opposite trends to model performance as heterogeneity varies. Specifically, with greater heterogeneity (*e.g.*, $\alpha = 0.1$), the model's pseudo-labeling ability is very limited, so including low-confidence samples may degrade performance due to a larger number of wrong pseudo-labels. But, when the labeling ability of the model is more reliable under lower heterogeneity (*e.g.*, $\alpha = 1$ or 10), simply discarding them may also miss a lot of correctly-labeled samples, leading to inferior performance. In short, the performance of the two methods is inconsistent across varying levels of data heterogeneity, and neither method shows clear superiority over the other. For Question ②, as illustrated in Fig. 1(c), the averaging-based parameter aggregation may deviate from the global category space due to the distribution discrepancy across clients.

To tackle the problems of FSSL with the above observations, we propose a new method called ProxyFL (**Proxy**-Guided **F**ederated Semi-Supervised **L**earning), **leveraging a unified proxy to simultaneously mitigate both internal and external heterogeneity**. *I.e.*, we consider the learnable weights of the classifier as proxy to model the category distribution both locally and globally. Proxy does not compromise data privacy or bring extra communication costs since the proxy itself is part of the model parameters in FL. Firstly, we introduce a **G**lobal **P**roxy-**T**uning (GPT) mechanism. This approach explicitly defines a global optimization objective to fit the category distribution across clients, mitigating distribution shift from external heterogeneity. Secondly, to compensate for the scarcity of local data, we incorporate low-confidence unlabeled samples via a dynamic **I**ndecisive-**C**ategories **P**roxy **L**earning (ICPL) mechanism. For each low-confidence sample, we propose an *indecisive-categories set* to represent its several possible categories instead of a single pseudo-label; For high-confidence unlabeled samples or labeled samples, we utilize the *pseudo-label* or *ground-truth*, respectively. Then we propose a relationship pool between unlabeled and labeled samples, and effectively train all samples based on the pool to mitigate internal heterogeneity. Experiments show that ProxyFL can significantly boost the performance and convergence of the FSSL model.

The main contributions of our paper are summarized as follows: ① To our best knowledge, this paper is the first to propose a unified proxy to mitigate both internal and external heterogeneity in FSSL. Note that our proxy does not compromise data privacy or bring extra communication costs. ② This paper proposes an FSSL method, ProxyFL, that can not only reduce the bias of averaging-based global parameters via an explicit optimization objective, but also precisely build the category relationship between all samples to facilitate more data participation, without compromising incorrect pseudo-labels. ③ This paper outperforms existing FSSL methods across multiple datasets and provides comprehensive experimental results. Our empirical & theoretical analysis also demonstrate our effectiveness and convergence under different levels of heterogeneity.

## 2 PROBLEM STATEMENT

This paper focuses on Federated Semi-Supervised Learning (FSSL) with both external and internal data heterogeneity. Specifically, we assume that a federation system $\mathbb{C}$ consists of $K$ clients, denoted

as $\mathbb{C} = \{\mathcal{C}_1, \ldots, \mathcal{C}_K\}$. Each client $\mathcal{C}_k$ maintains a private partially-labeled dataset $\mathcal{D}_k$, including labeled samples $\mathcal{D}_k^s = \{\mathbf{x}_{k,i}, \mathbf{y}_{k,i}\}_{i=1}^{N_k^s}$ and unlabeled samples $\mathcal{D}_k^u = \{\mathbf{u}_{k,i}\}_{i=1}^{N_k^u}$, where $N_k^s \ll N_k^u$. For each $\mathcal{C}_k$, its local model is parameterized by $\mathbf{\Theta}_k$, which comprises a feature extractor $f_k$ parameterized by $\theta_k$, projecting local data $\mathbf{x} \in \mathbb{R}^D$ to an embedding space $\mathbb{R}^d$, and a classifier $h_k$ parameterized by $\omega_k$, mapping the embedding space to category space $\mathbb{R}^C$, where $C$ indicates total category number. *I.e.*, $\mathbf{\Theta}_k = \theta_k \cup \omega_k$. Let $\mathcal{P}(\mathbf{Y})$ represent label distribution, and we formally define data heterogeneity in FSSL as follows:

**Definition 1 (External Heterogeneity in FSSL)** *External heterogeneity refers to the distribution discrepancy between $\mathcal{D}_k$ across different clients $\{\mathcal{C}_1, \ldots, \mathcal{C}_K\}$, i.e., for any two different clients $\mathcal{C}_{k_1}$ and $\mathcal{C}_{k_2}$, $\mathcal{P}_{k_1}(\mathbf{Y}) \neq \mathcal{P}_{k_2}(\mathbf{Y})$.*

**Definition 2 (Internal Heterogeneity in FSSL)** *Internal heterogeneity exists within local clients, embodied in: (1) class imbalance, arising from unequal sample sizes across different categories within client $\mathcal{C}_k$, i.e., for any two categories $c_1$ and $c_2$, $\mathcal{P}_k(\mathbf{Y}(c_1)) \neq \mathcal{P}_k(\mathbf{Y}(c_2))$; (2) distribution imbalance between labeled and unlabeled data, denoted as $\mathcal{P}_k^s(\mathbf{Y}) \neq \mathcal{P}_k^u(\mathbf{Y})$.*

The objective of FSSL is to train a shared global model parameterized by $\mathbf{\Theta}_{\mathcal{G}}$. During each communication round, a subset of online clients $\mathbb{C}_M \subseteq \mathbb{C}$ is randomly selected for local training (Liu et al. (2025)). On the central server, FSSL methods typically aggregate the uploaded local parameters $\{\mathbf{\Theta}_m\}_{\mathcal{C}_m \in \mathbb{C}_M}$ as the global parameters $\mathbf{\Theta}_{\mathcal{G}} = \sum_{\mathcal{C}_m \in \mathbb{C}_M} \gamma_m \mathbf{\Theta}_m$, where the aggregation weight $\gamma_m$ of $\mathcal{C}_m$ is empirically set by the proportion of its local dataset size relative to the total samples across all participating clients.

## 3 METHOD

### 3.1 PRELIMINARY STUDY

In this study, our goal is to simultaneously tackle both internal and external heterogeneity in FSSL. We conduct some exploratory experiments shown in Fig. 2. First, to explore **external heterogeneity**, we attempt to model the global category distribution under FedSGD on the central server. As shown in Yao et al. (2022), the weight parameters of the network classifier have a certain ability to differentiate categories. Thus, we extract the weight parameters $\{\omega_m\}_{\mathcal{C}_m \in \mathbb{C}_M}$, $\omega_m \in \mathbb{R}^{C \times d}$ of the uploaded local classifiers and slice them by class to generate a t-SNE plot, thereby visually showing the global category distribution across clients. As shown in Fig. 2(a), each pentagram represents the centroid of one category cluster, *e.g.*, the simple average of weight parameters $\{\omega_m^c\}_{\mathcal{C}_m \in \mathbb{C}_M}$ for the $c$-th category. We observe that directly using centroids may not accurately fit the global category distribution across clients. Due to data heterogeneity, the category distribution of some clients exhibits significant discrepancies from others, causing some points to be outliers, *e.g., the red and blue categories* in the upper of Fig. 2(a) with outliers pointed out. The simple-averaging method (pentagram) is affected by these outliers, positioning the centroid outside most points of the cluster for that category. So this method is hard to fit the real global distribution of per-class classifier weights across clients. We summarize as follows:

**Observation 1** *Simply averaging classifier weights is prone to skew towards outliers, thus failing to effectively capture the global category distribution across clients.*

For **internal heterogeneity**, most FSSL methods follow FixMatch (Sohn et al. (2020)) to assign pseudo-label only to high-confident unlabeled samples (*i.e.*, their prediction scores exceed a predefined threshold $\tau$), while excluding low-confident ones from training. This allows the model to heavily rely on limited easy-to-judge unlabeled samples and exacerbates internal heterogeneity. We examine internal heterogeneity in FSSL using FedSGD (McMahan et al. (2017)) in Fig. 2(b-c) and observe that:

**Observation 2** *As data heterogeneity increases, more unlabeled samples will be excluded from local training; Appropriately including the discarded samples could improve test performance.*

Therefore, it is crucial to effectively include these discarded samples into training for mitigating internal heterogeneity. Based on the above observations, it is expected to address the data hetero-

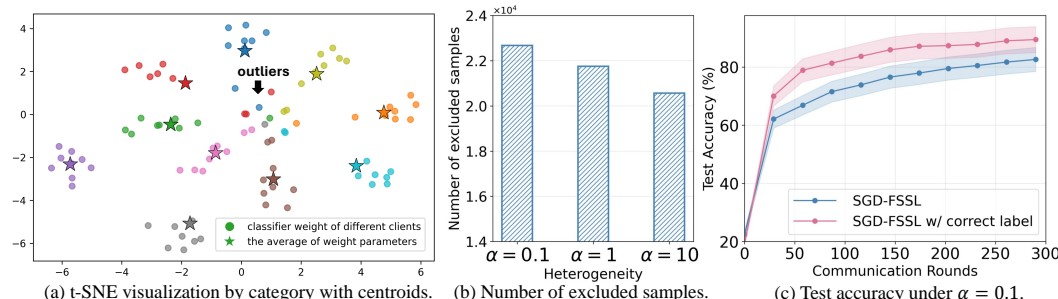

Figure 2: (a) t-SNE visualization of **classifier weights** from local clients (circle) during the initial round. Different colors means different categories. Pentagram denotes the simple average of weight parameters by category. (b) **Number of excluded unlabeled samples** under different levels of heterogeneity. As heterogeneity increases, more unlabeled samples are excluded from training. (c) **Test accuracy curves** of 'SGD-FSSL' and 'SGD-FSSL *w/* correct label' under $\alpha = 0.1$, where the latter denotes incorporating the excluded samples into training with GT labels. We find that those excluded samples have the potential to improve performance.

geneity in FSSL from two perspectives, *i.e.*, how to fit a global category distribution robustly against outliers (for external), and how to effectively leverage low-confidence unlabeled samples (for internal). To this end, based upon the learnable weights of the model classifier, we propose G̲lobal P̲roxy T̲uning (GPT) and I̲ndecisive-C̲ategories P̲roxy L̲earning (ICPL), globally improving global category distribution in a learnable manner, and locally enhancing unsupervised data utilization while mitigating the impact of incorrect pseudo-labels.

### 3.2 PROXY-GUIDED FEDERATED SEMI-SUPERVISED LEARNING

The goal of FL is to train a shared global model with a well-separated category distribution. To this end, previous FL approaches (Tan et al. (2022); Huang et al. (2023)) typically employ class prototypes to refine the category distribution, where global prototypes serve as the global category representations to regularize the local distribution. However, the main drawbacks of prototypes in FL are two-fold: 1) local prototypes are derived from sample features, posing a potential risk of feature leakage when uploaded to the server and subsequently re-distributed to other clients; 2) local prototypes need to be uploaded as additional burdens to the server, introducing extra communication costs. Inspired by Yao et al. (2022), **we consider the learnable weights of model classifier as proxy for modeling category distribution instead of prototypes**. *I.e.*, for client $\mathcal{C}_m$, we define the proxy vectors $\mathbf{\Omega}_m$ as $\{\omega_m^c\}_{c=1}^C$ in final FC layer to represent the $c$-th category, where $\omega_m^c \in \mathbb{R}^d$. As illustrated in Sec. 3.1, the proxy exhibits certain category-discriminative ability, since it could determine the category of a sample based on its features. Moreover, compared to sample features, the proxy serves as a natural component of model parameters, which communicate between local clients and the server without raising privacy concerns and avoiding extra communication overheads.

#### 3.2.1 GLOBAL PROXY TUNING

In FSSL, external heterogeneity refers to the label distribution discrepancy across clients. To mitigate its impact, we propose to model the global category distribution $\mathcal{P}_{\mathcal{G}}(\mathbf{Y})$ on the central server via learning a set of global proxies $\mathbf{\Omega}_{\mathcal{G}} = \{\omega_{\mathcal{G}}^c\}_{c=1}^C$, called G̲lobal P̲roxy T̲uning (GPT). In each communication round, the server receives model parameters from each client, thus a straight-forward idea to obtain $\mathbf{\Omega}_{\mathcal{G}}$ is $\overline{\mathbf{\Omega}}_{\mathcal{G}} = \{\sum_{\mathcal{C}_m \in \mathbb{C}_M} \gamma_m \omega_m^c\}_{c=1}^C$, where $C$ denotes the total category number, $\overline{\mathbf{\Omega}}_{\mathcal{G}}$ means the average operation, $\gamma_m$ denotes the aggregation weight of $\mathcal{C}_m$ based on the amount of data like FedAvg (McMahan et al. (2017)), and $\mathbb{C}_M$ is a subset of all clients $\mathbb{C}$ as Sec. 2. However, as summarized in Observation 1, simply averaging the local proxies is prone to be affected by the outliers such that the centroids show the discrepancy from the distribution of local proxies. Therefore, we propose to first initialize the global proxies $\mathbf{\Omega}_{\mathcal{G}}$ with $\overline{\mathbf{\Omega}}_{\mathcal{G}}$ and further fine-tune $\mathbf{\Omega}_{\mathcal{G}}$ by leveraging the off-the-shelf uploaded local proxies $\{\omega_m^c\}_{c=1}^C$, $\forall \mathcal{C}_m \in \mathbb{C}_M$ on the server. More concretely, for the global proxy $\mathbf{\Omega}_{\mathcal{G}}^c$ of category $c$, our objective is to pull it closer to all local proxies belonging to category $c$ and push it away from local proxies of other classes. The training objective of $\mathbf{\Omega}_{\mathcal{G}}$ can be

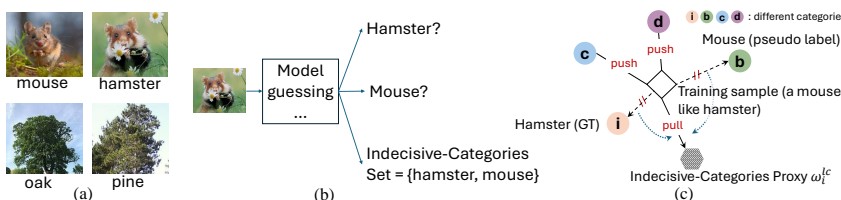

Figure 3: (a) Examples of the *indecisive-categories* in two groups. (b-c) Illustration of our Indecisive-Categories Proxy Learning (ICPL).

formulated as:

$$\min_{\mathbf{\Omega}_{\mathcal{G}}} \sum_{c=1}^{C} \sum_{m=1}^{M} \left[ \phi(\mathbf{\Omega}_{\mathcal{G}}^{c}, \omega_m^c) - \sum_{c'=1, c' \neq c}^{C} \phi(\mathbf{\Omega}_{\mathcal{G}}^{c}, \omega_m^{c'}) \right], \tag{1}$$

where $\phi(\cdot, \cdot)$ refers to the distance metric. Given the training objective, we formulate the loss function for Global Proxy Tuning (GPT) as follows:

$$\mathcal{L}_{GPT} = \sum_{c=1}^{C} \sum_{m=1}^{M} -log \frac{e^{-\phi(\mathbf{\Omega}_{\mathcal{G}}^{c}, \omega_m^c)}}{e^{-\phi(\mathbf{\Omega}_{\mathcal{G}}^{c}, \omega_m^c)} + \sum_{c'=1, c' \neq c}^{C} e^{-\phi(\mathbf{\Omega}_{\mathcal{G}}^{c}, \omega_m^{c'})}}. \tag{2}$$

The entire learning process of GPT module is conducted on the server. Then, the well-optimized global proxies $\mathbf{\Omega}_{\mathcal{G}}$ are sent to $M$ clients together with other global parameters, as the parametric initialization for the next round of local training.

### 3.2.2 INDECISIVE-CATEGORIES PROXY LEARNING

In FSSL, internal heterogeneity exists not only across different categories but also between labeled and unlabeled samples (as discussed in Definition 2). Current FSSL approaches (Jeong et al. (2020); Cho et al. (2023); Zhu et al. (2024)) either empirically set a fixed threshold (*e.g.*, 0.95 or 0.85) or design a dynamic threshold (Bai et al. (2024)) to filter out low-confidence unlabeled samples. These samples are directly excluded from local training. As noted in Observation 2, internal heterogeneity degrades model performance due to limited data participation. To this end, we propose Indecisive-Categories Proxy Learning (ICPL) to incorporate low-confidence samples into training. Specifically, for an unlabeled sample $\mathbf{u}_i$ [1] from $\mathcal{D}_k^u$ in client $\mathcal{C}_k$, its local feature $z_i$, local logits $\tilde{\mathbf{y}}_i$ and global logits $\overline{\mathbf{y}}_i$ via $\mathbf{\Theta}_g$ can be calculated as:

$$z_i = f_k\left(\mathcal{T}_w(\mathbf{u}_i)\theta_k\right), \ \tilde{\mathbf{y}}_i = h_k\left(z_i; \omega_k\right), \ \overline{\mathbf{y}}_i = h_{\mathcal{G}}((f_{\mathcal{G}}(\mathcal{T}_w(\mathbf{u}_i); \theta_{\mathcal{G}})); \mathbf{\Omega}_{\mathcal{G}}), \tag{3}$$

where $\mathcal{T}_w(\mathbf{u}_i)$ denotes the weakly-augmented version of $\mathbf{u}_i$ and $\tilde{\mathbf{y}}_i, \overline{\mathbf{y}}_i \in \mathbb{R}^C$. If $\max(\overline{\mathbf{y}}_i) > \tau$, $\mathbf{u}_i$ is a high-confidence sample (denoted as $\mathbf{u}_i^{hc}$) and its pseudo-label $\hat{\mathbf{y}}_i = \arg\max(\overline{\mathbf{y}}_i)$[2]; otherwise, $\mathbf{u}_i$ is regarded as a low-confidence sample (denoted as $\mathbf{u}_i^{lc}$). In this case, it may affect model performance to directly assign category via $\arg\max$ due to potentially incorrect pseudo-labels. Prior study (Chen et al. (2022)) has shown the effectiveness of assigning more than one category labels to low-confidence unlabeled samples. As illustrated in Fig. 3(a), datasets often contain some fine-grained classes that are difficult to distinguish, especially when two classes belong to the same superclass, *e.g.*, mouse & hamster, oak & pine. For a mouse-like hamster image, the model is uncertain whether the object is a mouse or a hamster during the pseudo-labeling process, but typically will not associate it with irrelevant categories such as a truck. Thus, for $\mathbf{u}_i^{lc}$, we define the several categories among which the model hesitates as its *indecisive-categories set* $\xi_i$, *e.g.*, {*mouse, hamster*} in Fig. 3(b). To design $\xi$, we are inspired by the category imbalance issue in FSSL, *i.e.*, leveraging the per-category number of labeled data across clients to simulate a global category prior $\mathcal{P}'_{\mathcal{G}}(\mathbf{Y})$ and then define the *indecisive-categories set* $\xi_i$ for a low-confidence sample $\mathbf{u}_i^{lc}$ based on the prior:

$$\mathcal{P}'_{\mathcal{G}}(\mathbf{Y}) = \{\sum_{k=1}^{K} N_k^s(c)\}_{c=1}^{C}; \ \xi_i = \{c \mid c \in [1, C] \land \tilde{\mathbf{y}}_i(c) > \mathcal{P}'_{\mathcal{G}}(\mathbf{Y}(c))\}, \tag{4}$$

---

[1] Note that in this section, we drop the client subscript $k$ in data samples for symbolic simplicity.

[2] To be clear, we re-claim that for an unlabeled sample $\mathbf{u}_i$, $\tilde{\mathbf{y}}_i$ and $\overline{\mathbf{y}}_i$ respectively denotes the local logits from local model and the global logits from global model, while $\hat{\mathbf{y}}_i$ represents the pseudo-labels.

where $N_k^s$ denotes the number of labeled samples of client $\mathcal{C}_k$. So for a low-confidence sample $\mathbf{u}_i^{lc}$, any category $c$ with corresponding logits $\tilde{\mathbf{y}}_i(c)$ exceeding $\mathcal{P}'_\mathcal{G}(\mathbf{Y}(c))$ will be considered as *indecisive category*. $\mathcal{P}'_\mathcal{G}(\mathbf{Y})$ acts as a dynamic threshold for different classes by setting a higher threshold for the majority classes and lower for the minority. Then, we leverage Contrastive Learning (CL) to address internal heterogeneity (Definition 2) in FSSL since CL directly reflects the relationships between samples. To this end, we first construct a *Positive-Negative Proxy Pool* for all unlabeled data in a batch and we remark the pool as follows:

**Remark 1 (Positive-Negative Proxy Pool)** *For an unlabeled batch $\mathcal{B}^u$ from $\mathcal{D}_k^u$, we construct the Positive-Negative Proxy Pool without excluding any of the samples. To be clear, we re-view several notations: for any $\mathbf{u}_i$ from $\mathcal{D}_k^u$, its local feature $z_i$, local logits $\tilde{\mathbf{y}}_i$ and global logits $\overline{\mathbf{y}}_i$ are calculated as Eq. 3. If $\max(\overline{\mathbf{y}}_i) > \tau$, $\mathbf{u}_i$ is a high-confidence sample with it pseudo-label $\hat{\mathbf{y}}_i = \arg\max(\overline{\mathbf{y}}_i)$; otherwise, $\mathbf{u}_i$ is a low-confidence sample with its indecisive-categories set $\xi_i$. Thus, for a high-confidence sample $\mathbf{u}_i^{hc}$, its positive proxy $\omega_i^{hc}$ is the weight $\omega_k^{\hat{\mathbf{y}}_i}$ of argmax-derived category $\hat{\mathbf{y}}_i$, i.e., $\omega_i^{hc} = \omega_k^{\hat{\mathbf{y}}_i}$. Its negative proxies are a set of feature vectors $\mathcal{R}_i^{hc} = \left\{z_j \big| z_j = f_k(\mathcal{T}_w(\mathbf{u}_j); \theta_k) \wedge \mathbf{u}_j \in \varphi_i^{hc}\right\}$, where $\varphi_i^{hc}$ is a set of unlabeled samples meeting certain conditions:*

$$\varphi_i^{hc} = \{\mathbf{u}_j \mid \mathbf{u}_j \in \mathcal{B}^u \wedge ((\max(\tilde{\mathbf{y}}_j) > \tau \wedge \hat{\mathbf{y}}_i \neq \hat{\mathbf{y}}_j) \vee (\max(\tilde{\mathbf{y}}_j) \leq \tau \wedge \hat{\mathbf{y}}_i \notin \xi_j))\}. \quad (5)$$

**For a low-confidence sample $\mathbf{u}_i^{lc}$**, *its positive proxy is derived from $\xi_i$, i.e.,*

$$\omega_i^{lc} = \sum_{\forall c' \in \xi_i} \tilde{\mathbf{y}}_i(c') \times \omega_k^{c'}, \quad (6)$$

*where $\omega_i^{lc}$ is designed to the weighted sum of proxy weights, based on the categories within the indecisive-categories set $\xi_i$. $\tilde{\mathbf{y}}_i(c')$ denotes the prediction score of $\mathbf{u}_i$ for the $c'$-th class. The negative proxy of $\mathbf{u}_i^{lc}$ is also a set $\mathcal{R}_i^{lc} = \left\{z_j \big| z_j = f_k(\mathcal{T}_w(\mathbf{u}_j); \theta_k) \wedge \mathbf{u}_j \in \varphi_i^{lc}\right\}$, where $\varphi_i^{lc}$ is a set of unlabeled samples meeting certain conditions:*

$$\varphi_i^{lc} = \{\mathbf{u}_j \mid \mathbf{u}_j \in \mathcal{B}^u \wedge ((\max(\tilde{\mathbf{y}}_j) > \tau \wedge \hat{\mathbf{y}}_j \notin \xi_i) \vee (\max(\tilde{\mathbf{y}}_j) \leq \tau \wedge \xi_i \cap \xi_j = \emptyset))\}. \quad (7)$$

According to the above equations, we establish category relationships among all unlabeled samples while reducing potential errors of pseudo-labels. The objective of ICPL can be formulated as:

$$\mathcal{L}_{ICPL} = -\left[\frac{1}{|\mathcal{B}^{u,hc}|} \sum_{i=1}^{|\mathcal{B}^{u,hc}|} \log \frac{e^{z_i \cdot \omega_i^{hc}}}{e^{z_i \cdot \omega_i^{hc}} + \sum_{z_j \in \mathcal{R}_i^{hc}} e^{z_i \cdot z_j}} + \frac{1}{|\mathcal{B}^{u,lc}|} \sum_{i=1}^{|\mathcal{B}^{u,lc}|} \log \frac{e^{z_i \cdot \omega_i^{lc}}}{e^{z_i \cdot \omega_i^{lc}} + \sum_{z_j \in \mathcal{R}_i^{lc}} e^{z_i \cdot z_j}}\right]. \quad (8)$$

where $|\mathcal{B}^{u,hc}|$ and $|\mathcal{B}^{u,lc}|$ denotes the number of high-confidence samples and low-confidence samples in batch $\mathcal{B}^u$, respectively. Remark 1's discussion about $\mathbf{u}^{lc}$ is further visualized in Fig. 3(c) for better understanding. **Given a low-confidence sample $\mathbf{u}_i^{lc}$**, we take a weighted average of category proxies from $\xi_i$ as its positive proxy $\omega_i^{lc}$, and pull $\mathbf{u}_i^{lc}$ closer to $\omega_i^{lc}$ to prevent $\mathbf{u}_i^{lc}$ biasing towards potentially-incorrect class from pseudo-label; Concurrently, two types of negative samples are selected to be pushed away: ① high-confidence samples in $\mathcal{B}^u$ whose top predicted class is not in $\xi_i$. ② other low-confidence samples in $\mathcal{B}^u$ whose indecisive-categories set do not overlap with $\xi_i$. **This strategy avoids incorrect sample-to-sample relationships as much as possible.**

As illustrated in Definition 2, we know that internal heterogeneity exists not only in class imbalance but also in **distribution imbalance between labeled and unlabeled data**. Thus, we expand the negative proxy set of unlabeled samples $\mathcal{R}^{hc}$ and $\mathcal{R}^{lc}$ through **including labeled samples into the pool.** For one labeled data $\mathbf{x}_j \in \mathcal{D}_k^s$, the ground-truth label $\mathbf{y}_j$ can directly represent its category. Thus we re-define $\mathcal{R}_i^{hc}$ and $\mathcal{R}_i^{lc}$ for $\mathbf{u}_i$ as follows:

**Remark 2** *Assume that there is a batch $\mathcal{B}$, consisting of labeled batch $\mathcal{B}^s$ and unlabeled batch $\mathcal{B}^u$. For a high-confidence unlabeled sample $\mathbf{u}_i^{hc}$, its negative proxies $\mathcal{R}_i^{hc}$ set can be expanded as:*

$$\dot{\mathcal{R}}_i^{hc} = \mathcal{R}_i^{hc} \cup \{z_j | z_j = f_k(\mathbf{x}_j; \theta_k) \wedge \mathbf{x}_j \in \mathcal{B}^s \wedge \mathbf{y}_j \neq \hat{\mathbf{y}}_i\}. \quad (9)$$

*Likewise, for a low-confidence unlabeled sample $\mathbf{u}_i^{lc}$, $\mathcal{R}_i^{lc}$ will be:*

$$\dot{\mathcal{R}}_i^{lc} = \mathcal{R}_i^{lc} \cup \{z_j | z_j = f_k(\mathbf{x}_j; \theta_k) \wedge \mathbf{x}_j \in \mathcal{B}^s \wedge \mathbf{y}_j \notin \xi_i\}. \quad (10)$$

After expanding $\mathcal{R}_i^{hc}$ and $\mathcal{R}_i^{lc}$, we compute $\mathcal{L}_{ICPL}$ (Eq. 8) by replacing $\mathcal{R}_i^{hc}, \mathcal{R}_i^{lc}$ with $\dot{\mathcal{R}}_i^{hc}, \dot{\mathcal{R}}_i^{lc}$.

### 3.3 Loss Functions

We summarize the entire training process of our method. In local training, we follow previous studies (Li et al. (2023)) to assign *ground-truth* $\mathbf{y}$ for labeled data and *pseudo-label* $\hat{\mathbf{y}}$ for high-confidence unlabeled data, respectively. Following SAGE (Liu et al. (2025)), the local losses are:

$$\mathcal{L}_u = \frac{1}{|\mathcal{B}^{u,hc}|} \sum_{i=1}^{|\mathcal{B}^{u,hc}|} \mathbf{KL}(h_k(f_k(\mathcal{T}_s(\mathbf{u}_i);\theta_k);\omega_k) \parallel \hat{\mathbf{y}}_i), \mathcal{L}_s = \frac{1}{|\mathcal{B}^s|} \sum_{i=1}^{|\mathcal{B}^s|} \mathcal{L}_{CE}((h_k(f_k(\mathbf{x}_i;\theta_k);\omega_k), \mathbf{y}_i),$$

$$(11)$$

where $\mathbf{KL}$ denotes Kullback-Leibler divergence loss, $\mathcal{T}_s(\cdot)$ denotes the strong-augmentation, $\mathcal{L}_{CE}$ denotes cross-entropy loss. Based on this, our approach effectively leverages **a unified proxy** to locally incorporate low-confidence unlabeled samples $\mathbf{u}^{lc}$ for training via $\mathcal{L}_{ICPL}$ and improve the global category distribution via $\mathcal{L}_{GPT}$ on the server. Thus, our final total objective is:

$$\mathcal{L} = \underbrace{\mathcal{L}_s + \alpha\mathcal{L}_u + \beta\mathcal{L}_{ICPL}}_{\text{local}} + \underbrace{\mathcal{L}_{GPT}}_{\text{global}},$$

$$(12)$$

where $\alpha, \beta$ are empirically set to 1.

## 4 Experiments

### 4.1 Experimental Setup

**Datasets.** We strictly follow the FSSL experimental setting of SAGE method (Liu et al. (2025)). Our method is evaluated on the CIFAR10, CIFAR-100, SVHN, and CINIC-10 datasets (Darlow et al. (2018); Krizhevsky et al. (2009); Netzer et al. (2011)). We partition the labeled and unlabeled samples per category with label proportions of 10% for each dataset. Following previous FSSL works (Zhu et al. (2024); Bai et al. (2024); Cho et al. (2023)), we simulate internal and external heterogeneity by sampling labeled & unlabeled data from a Dirichlet distribution $Dir(\alpha)$ and allocate them to local clients with three levels of $Dir(\alpha)$: $\alpha = \{0.1, 0.5, 1\}$. The smaller $\alpha$, the higher FL data heterogeneity. We visualize the specific data distribution in Fig. 4(a).

**Implementation Details** Following SAGE (Liu et al. (2025)), we configure 20 clients for all settings, with 8 clients randomly sampled each round to participate in the federated training. ResNet-8 (He et al. (2016)) serves as the local backbone, with the number of local epochs set to 5, local learning rate set to 0.1 and the confidence threshold $\tau$ for pseudo-labeling set to 0.95. For global proxy tuning process, the learning rate is 0.005, and the number of server epochs is set to 10 for CIFAR-100 and 100 for the other datasets. Unless otherwise specified, the experimental setup of ProxyFL is consistent with SAGE (Liu et al. (2025)).

### 4.2 Performance Comparison

Tab. 1 reports the overall results of our ProxyFL and other state-of-the-art methods across different datasets under different Non-IID (**Non-I**ndependent and **I**dentical **D**istribution) scenarios with 10% label. We compare the following methods in our experiments like SAGE (Liu et al. (2025)): ① **FL methods** For FedAvg (McMahan et al. (2017)) and FedProx (Li et al. (2020)), all clients are trained via supervised federated learning only on labeled data part; For FedAvg-SL, local data are all fully-labeled datasets, which denotes the ideal upper-bound based on standard fully-supervised FedAvg. ② **Vanilla combinations (FL + SSL methods)** Here, each method denotes a simple combination of SSL methods and FL methods. Note that FixMatch-LPL and FixMatch-GPL are both FixMatch-based frameworks, but pseudo-labels (PL) are derived from different models, *i.e.*, local model for LPL and global model for GPL, respectively. ③ **FSSL methods** We compare ProxyFL with previous state-of-the-art federated semi-supervised learning (FSSL) methods, including FedMatch (Jeong et al. (2020)), FedLabel (Cho et al. (2023)), FedLoke (Zhang et al. (2023)), FedDure (Bai et al. (2024)), FedDB (Zhu et al. (2024)) and SAGE (Liu et al. (2025)).

In Tab. 1, ProxyFL achieves state-of-the-art performances on all datasets with significant improvements under different levels of data heterogeneity $\alpha$. **To the best of our knowledge, we are the**

| Methods | CIFAR-10 | | | CIFAR-100 | | | SVHN | | | CINIC-10 | | |
|---|---|---|---|---|---|---|---|---|---|---|---|---|
| | $\alpha=0.1$ | $\alpha=0.5$ | $\alpha=1$ | $\alpha=0.1$ | $\alpha=0.5$ | $\alpha=1$ | $\alpha=0.1$ | $\alpha=0.5$ | $\alpha=1$ | $\alpha=0.1$ | $\alpha=0.5$ | $\alpha=1$ |
| **FL Methods** | | | | | | | | | | | | |
| FedAvg | 69.60 | 68.88 | 69.39 | 34.08 | 33.21 | 35.31 | 82.40 | 83.40 | 78.60 | 57.17 | 60.09 | 61.54 |
| FedProx | 68.58 | 69.53 | 68.00 | 34.20 | 34.07 | 34.88 | 81.67 | 83.77 | 83.77 | 58.05 | 60.71 | 62.82 |
| FedAvg-SL | 90.46 | 91.24 | 91.32 | 67.98 | 68.83 | 69.10 | 94.11 | 94.41 | 94.40 | 77.82 | 80.42 | 81.29 |
| **FL+SSL Methods** | | | | | | | | | | | | |
| FixMatch-LPL | 82.98 | 84.36 | 84.69 | 49.32 | 49.67 | 49.55 | 89.68 | 91.33 | 91.91 | 68.02 | 70.67 | 72.69 |
| FixMatch-GPL | 84.56 | 86.05 | 86.66 | 48.96 | 51.80 | 52.19 | 90.50 | 91.94 | 92.31 | 71.67 | 73.26 | 74.80 |
| FedProx+FixMatch | 84.60 | 85.49 | 86.95 | 48.42 | 48.51 | 49.33 | 90.46 | 91.36 | 91.25 | 68.62 | 70.67 | 72.69 |
| FedAvg+FlexMatch | 84.21 | 86.00 | 86.57 | 49.91 | 51.39 | 51.79 | 52.58 | 55.59 | 60.50 | 69.20 | 71.87 | 73.42 |
| **FSSL Methods** | | | | | | | | | | | | |
| FedMatch | 75.35 | 77.86 | 78.00 | 32.23 | 31.49 | 35.75 | 88.63 | 89.20 | 89.23 | 51.94 | 56.27 | 70.22 |
| FedLabel | 62.85 | 79.46 | 79.17 | 50.88 | 52.21 | 52.38 | 89.31 | 91.51 | 91.16 | 67.64 | 70.56 | 72.80 |
| FedLoke | 83.32 | 82.22 | 81.87 | 39.29 | 40.46 | 39.96 | 89.94 | 90.00 | 89.45 | 59.03 | 61.60 | 63.21 |
| FedDure | 84.60 | 85.88 | 87.34 | 48.27 | 51.09 | 50.79 | 92.87 | 93.49 | 94.19 | 70.86 | 73.37 | 74.89 |
| FedDB | 83.99 | 85.28 | 87.49 | 48.43 | 50.11 | 51.55 | 92.56 | 93.00 | 93.14 | 69.44 | 72.60 | 73.61 |
| SAGE | 87.05 | 88.05 | 89.08 | 54.18 | 55.82 | 56.06 | 93.85 | 94.27 | 94.65 | 74.59 | 75.74 | 76.68 |
| **ProxyFL (ours)** | **88.56** | **90.00** | **89.96** | **57.50** | **58.75** | **58.24** | **95.09** | **95.18** | **95.26** | **77.98** | **78.96** | **79.59** |
| | ↑1.51 | ↑1.95 | ↑0.88 | ↑3.32 | ↑2.93 | ↑2.18 | ↑1.24 | ↑0.91 | ↑0.61 | ↑3.39 | ↑3.22 | ↑2.91 |

Table 1: Experimental results on CIFAR-10, CIFAR-100, SVHN and CINIC-10 under 10% label. Bold text indicates the best result, and the last row presents the improvement of ProxyFL over the second best method.

| GPT | ICPL | CIFAR10 | | | CIFAR100 | | | CINIC10 | | | SVHN | | |
|---|---|---|---|---|---|---|---|---|---|---|---|---|---|
| | | $\alpha=0.1$ | $\alpha=0.5$ | $\alpha=1$ | $\alpha=0.1$ | $\alpha=0.5$ | $\alpha=1$ | $\alpha=0.1$ | $\alpha=0.5$ | $\alpha=1$ | $\alpha=0.1$ | $\alpha=0.5$ | $\alpha=1$ |
| | | 84.56 | 86.05 | 86.66 | 48.96 | 51.80 | 52.19 | 90.50 | 91.94 | 92.31 | 71.67 | 73.26 | 74.80 |
| ✓ | | 87.59 | 89.23 | 89.71 | 54.86 | 56.58 | 57.09 | 94.29 | 94.49 | 94.53 | 77.15 | 79.03 | 79.31 |
| | ✓ | 87.81 | 89.58 | 89.66 | 57.21 | 57.98 | 57.74 | 94.82 | 94.69 | 95.15 | 77.80 | 78.04 | 78.57 |
| ✓ | ✓ | **88.56** | **90.00** | **89.96** | **57.50** | **58.75** | **58.24** | **95.09** | **95.18** | **95.26** | **77.98** | **78.96** | **79.59** |

Table 2: Module ablation studies on GPT and ICPL of our method.

**first in FSSL to propose category proxy for mitigating both internal and external heterogeneity.** Notably, our ProxyFL even achieves comparable performance to that of FedAvg-SL on SVHN dataset and CINIC-10 when $\alpha = 0.1$. We attribute this improvement to the generalization of the enhanced category distribution from our proxy-guided FSSL framework.

### 4.3 ABLATION STUDIES

In this section, we conduct an in-depth investigation to validate the contributions of our GPT and ICPL in ProxyFL.

**Effectiveness of Modules** We first validate the contributions of GPT and ICPL through ablation experiments and set FedAvg+FixMatch-GPL as our baseline model. We conduct ablation studies under different level of heterogeneity $\alpha = \{0.1, 0.5, 1\}$ to assess the effectiveness of each module. As shown in Tab. 2, In our framework, each module could individually enhance model performance and their combination of GPT & ICPL achieves the best results.

**Convergence Analysis** Fig. 4(b) and Tab. 3 demonstrate that ProxyFL substantially improves the convergence speed and test accuracy on the CIFAR-100 dataset. ProxyFL outperforms baseline and current FSSL methods, achieving better performance with greater communication efficiency. Most existing FSSL methods (Jeong et al. (2020); Cho et al. (2023); Zhu et al. (2024)) only retain high-confidence samples for training while discarding the low-confidence ones, leading to slower model convergence due to fewer training samples under non-IID scenarios. In contrast, ProxyFL effectively incorporates low-confidence samples and leverages category proxies to address both internal and external heterogeneity, thereby accelerating model convergence especially during the initial training stages. In addition, we also provide some theoretical proofs for convergence in Appendix C.

**Analysis of Global Proxy Tuning** We explore the effect of our GPT module by visualizing the proxy distribution across clients in a t-SNE plot. As observed in Fig. 4(c), the squares (*the proxies after tuning*) fit more accurately the proxy distribution across clients than the pentagram centroids (*the directly-averaging proxies*), showing better robustness to outliers. By explicitly defining an

| Acc. | 30% | | 40% | | 50% | |
|---|---|---|---|---|---|---|
| Method | Round↓ | Speedup↑ | Round↓ | Speedup↑ | Round↓ | Speedup↑ |
| LPL | 119 | ×1.00 | 242 | ×1.00 | 562 | ×1.00 |
| GPL | 114 | ×1.04 | 226 | ×1.07 | 524 | ×1.07 |
| FedLabel | 94 | ×1.27 | 175 | ×1.38 | 429 | ×1.31 |
| FedDB | 103 | ×1.16 | 206 | ×1.17 | - | - |
| FedDure | 114 | ×1.04 | 234 | ×1.03 | 542 | ×1.04 |
| SAGE | 60 | ×1.98 | 124 | ×1.95 | 267 | ×2.10 |
| **ProxyFL** | 45 | ×2.64 | 89 | ×2.72 | 177 | ×3.18 |

| Methods | CIFAR-10 | CIFAR-100 | SVHN | CINIC-10 |
|---|---|---|---|---|
| FedAvg-SL | 90.46 | 67.98 | 94.11 | 77.82 |
| GPL | 84.56 | 48.96 | 90.50 | 71.67 |
| GPL-ALL | 85.38 | 50.34 | 93.31 | 75.83 |
| LPL | 82.98 | 49.32 | 89.68 | 68.02 |
| LPL-ALL | 87.18 | 53.28 | 83.99 | 73.39 |
| ICPL-Top1 | 87.13 | 55.66 | 94.56 | 77.01 |
| ICPL-Top5 | 87.77 | 56.58 | 94.71 | 77.65 |
| **ICPL-$\mathcal{P}'_{\mathcal{G}}(\mathbf{Y})$** | 87.81 | 57.21 | 94.82 | 77.80 |

Table 3: Comparison of convergence rates between ProxyFL and other baselines with $\alpha = 0.1$ on CIFAR-100.

Table 4: Ablation of design choices between ICPL and other methods with $\alpha = 0.1$.

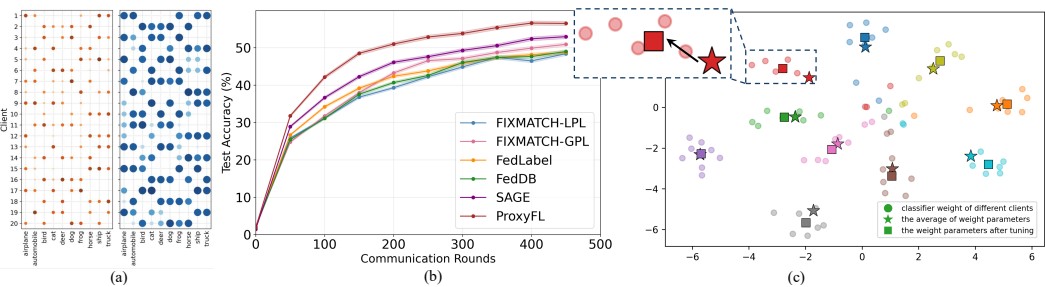

Figure 4: (a) Distribution of labeled and unlabeled data across clients under $\alpha = 0.1$ taking CIFAR-10 as an example. (b) Convergence curves of ProxyFL and other baselines on CIFAR-100 with $\alpha = 0.1$. (c) Distribution of global category proxies before-and after-tuning visualized in a t-SNE plot.

optimization objective for global category distribution, our method can fit the real global distribution of per-class classifier weights across clients.

**Analysis of Indecisive-Categories Proxy Learning** We analyze the effectiveness of ICPL. An intuitive idea to include low-confidence samples is to directly assign pseudo-labels for them like high-confidence samples, abbreviated as Fixmatch-LPL-ALL and Fixmatch-GPL-ALL. And FedAvg-SL, the standard fully-labeled FedAvg, serves as an upperbound with correct labels. As shown in Tab. 4, in most cases, directly including all unlabeled samples could bring slight improvements compared to simply-discarding, suggesting that low-confidence samples contain valuable information for training. So simply discarding them may exclude some correctly-labeled samples from training; But, directly including them sometimes leads to performance degradation, *e.g.*, LPL & LPL-ALL on SVHN dataset and underperforms FedAvg-SL due to incorrect pseudo-labels. Compared to discarding or directly including low-confidence samples, our proposed ICPL module achieves the best performance across all datasets by more accurately constructing the relationships between samples in the positive-negative pool of ICPL. Moreover, ICPL even reaches the performance of FedAvg-SL on certain datasets.

**Design of Indecisive-Categories Set $\xi$** We discuss the design of indecisive-categories set $\xi$ for low-confidence samples $\mathbf{u}^{1c}$. We compare our strategy of federated global prior $\mathcal{P}'_{\mathcal{G}}(\mathbf{Y})$ for $\xi$ with the commonly-used designs that selects the categories from Top-1 or Top-5 confidence scores. As shown in Tab. 4, our method consistently yields better performance than other designs, which validates the effectiveness of $\mathcal{P}'_{\mathcal{G}}(\mathbf{Y})$ that sets different thresholds for different categories.

## 5 CONCLUSION

Our paper presents a new Federated Semi-Supervised Learning (FSSL) method called ProxyFL, leveraging a unified proxy to simultaneously mitigate external and internal heterogeneity. We model the category distribution both locally and globally. Firstly, we define a global optimization objective to fit the category distribution across clients, mitigating distribution shift from external heterogeneity. Secondly, we incorporate low-confidence unlabeled samples via the proposed dynamic indecisive categories proxy learning mechanism to mitigate internal heterogeneity. Experiments show that ProxyFL can ignificantly boost the performance and convergence of the FSSL model.

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

## A  PSEUDO-CODE

To facilitate a clearer understanding of our approach, we present the pseudo-code of ProxyFL in Algorithm 1.

---

**Algorithm 1:** Proxy-Guided Federated Semi-Supervised Learning (ProxyFL)

---

**Input:** Set of clients $\mathbb{C}$; number of online clients $M$; number of communication rounds $T$; number of local epochs $E$; number of server epochs $Q$; weak augmentation $\mathcal{T}_w(\cdot)$; strong augmentation $\mathcal{T}_s(\cdot)$; confidence threshold $\tau$; local learning rate $\eta_l$; server learning rate $\eta_{\mathcal{G}}$; loss weights $\alpha, \beta$.

1 **ServerExecutes:**
2 Randomly initialize global model parameters $\boldsymbol{\Theta}_{\mathcal{G}}^0 = \{\theta_{\mathcal{G}}^0, \boldsymbol{\Omega}_{\mathcal{G}}^0\}$
3 **for** $t = 0$ **to** $T - 1$ **do**
4 $\quad$ Randomly select a subset of online clients $\mathbb{C}_M \subseteq \mathbb{C}$
5 $\quad$ **foreach** *client* $\mathcal{C}_m \in \mathbb{C}_M$ ***in parallel*** **do**
6 $\quad\quad$ $\boldsymbol{\Theta}_m^{t+1} \leftarrow$ **ClientUpdate** $(\boldsymbol{\Theta}_{\mathcal{G}}^t)$;
7 $\quad$ **end**
8 $\quad$ $\theta_{\mathcal{G}}^{t+1} \leftarrow \sum_{\mathcal{C}_m \in \mathbb{C}_M} \gamma_m \theta_m^{t+1}; \overline{\boldsymbol{\Omega}}_{\mathcal{G}} \leftarrow \sum_{\mathcal{C}_m \in \mathbb{C}_M} \gamma_m \omega_m^{t+1}$.
$\quad\quad$ // Global Proxy Tuning (GPT)
9 $\quad$ Initialize $\boldsymbol{\Omega}_{\mathcal{G}}$ by the average of local proxies: $\boldsymbol{\Omega}_{\mathcal{G}} \leftarrow \overline{\boldsymbol{\Omega}}_{\mathcal{G}}$.
10 $\quad$ **for** $e = 1$ **to** $Q$ **do**
11 $\quad\quad$ Compute $\mathcal{L}_{GPT}$ with local proxies $\{\omega_m^c\}_{c=1}^C, \forall \mathcal{C}_m \in \mathbb{C}_M$.
12 $\quad\quad$ Tune global proxies $\boldsymbol{\Omega}_{\mathcal{G}} \leftarrow \boldsymbol{\Omega}_{\mathcal{G}} - \eta_{\mathcal{G}} \nabla_{\boldsymbol{\Omega}_{\mathcal{G}}} \mathcal{L}_{GPT}$ in Eq. 2.
13 $\quad$ **end**
14 $\quad$ $\boldsymbol{\Omega}_{\mathcal{G}}^{t+1} \leftarrow \boldsymbol{\Omega}_{\mathcal{G}}$
15 **end**
16 **return** $\boldsymbol{\Theta}_{\mathcal{G}}^T = \theta_{\mathcal{G}}^T \cup \boldsymbol{\Omega}_{\mathcal{G}}^T$
17 **ClientUpdate** $(\boldsymbol{\Theta}_{\mathcal{G}}^t)$**:**
18 Initialize local model parameters $\boldsymbol{\Theta}_k^t = \theta_k^t \cup \omega_k^t$ via $\theta_k^t \leftarrow \theta_{\mathcal{G}}^t$ and $\omega_k^t \leftarrow \boldsymbol{\Omega}_{\mathcal{G}}^t$
19 **for** $e = 1$ **to** $E$ **do**
20 $\quad$ Compute supervised loss $\mathcal{L}_s$ as Eq. 11 for labeled samples $\mathbf{x} \in \mathcal{D}_k^s$;
21 $\quad$ Compute consistency loss $\mathcal{L}_u$ as Eq. 11 for high-confidence unlabeled samples $\mathbf{u}^{hc} \in \mathcal{D}_k^u$;
$\quad\quad$ // Indecisive-Categories Proxy Learning (ICPL)
22 $\quad$ Calculate $\mathcal{L}_{ICPL}$ for all unlabeled samples from $\mathcal{D}^u$ by constructing a positive-negative proxy pool (Remarks 1 & 2) and using Eq. 8.
23 $\quad$ $\mathcal{L}_{local} \leftarrow \mathcal{L}_s + \alpha \mathcal{L}_u + \beta \mathcal{L}_{ICPL}$.
24 $\quad$ $\boldsymbol{\Theta}_k \leftarrow \boldsymbol{\Theta}_k - \eta_l \nabla_{\boldsymbol{\Theta}_k} \mathcal{L}_{local}$.
25 **end**
26 **return** $\boldsymbol{\Theta}_k^{t+1} = \{\theta_k, \omega_k\}$

---

## B  RELATED WORK

### B.1  FEDERATED LEARNING

Federated learning (FL) is a distributed machine learning approach that focus on safeguarding data privacy. With the development of FedAvg (McMahan et al. (2017)) as the pioneer FL algorithm, researchers have delved into the study of FL. One of the most significant challenges is data heterogeneity, meaning that the distributions among different clients are non-i.i.d, *i.e.*, non-independent and identically distributed. FedProx (Li et al. (2020)) requires each client to regularize with the global model parameters during local training to prevent the impact of local bias. A large amount of works have been proposed to address this challenge with approaches such as additional data sharing, regularization, aggregation strategies, and personalization (Li et al. (2021); Tan et al. (2022); Li et al. (2024); Shi et al. (2025)). However, these fully-supervised FL approaches struggle to generalize well under the scenarios of annotation scarcity. To this end, Federated Semi-Supervised

Learning (FSSL) has emerged, enabling clients to train models leveraging both limited labeled data and a large amount of unlabeled data, thereby improving the performance of global model. Our work mainly targets on federated semi-supervised learning, where a small fraction of samples has ground-truth labels in each client (Bai et al. (2024); Liu et al. (2025)).

### B.2 SEMI-SUPERVISED LEARNING

Semi-Supervised Learning (SSL) aims to effectively leverage both limited labeled data and a large amount of unlabeled data to improve model performance. Two commonly-used strategies in SSL are consistency regularization and pseudo-labeling, respectively. Consistency regularization is based on the assumption that a model's prediction should remain consistent despite diverse perturbations to the inputs or model (Chen et al. (2021); Yun et al. (2019); Olsson et al. (2021)). Another common strategy is pseudo-labeling, which determines pseudo-labels for unlabeled samples based on the high-confidence predictions of the model trained by labeled data and filter high-confidence unlabeled samples as training samples (Sohn et al. (2020); Zhang et al. (2021); Wang et al. (2022)). However, pseudo-label generation in self-training based methods heavily depends on prediction confidence score, and if these methods are simply transferred to the field of FL, the number of local data will further decline due to the exclusion of low-confidence unlabeled samples.

### B.3 FEDERATED SEMI-SUPERVISED LEARNING

Federated Semi-Supervised Learning (FSSL) addresses the challenge of training models on decentralized data where labels are scarce. The field is often categorized into three distinct scenarios: ① Labels-at-Server, where clients only have unlabeled data and a central server holds labeled data (Diao et al. (2022); He et al. (2021); Jeong et al. (2020); Kim et al. (2023); Yang et al. (2024)). ② Label-at-All-Client, where every client has a small fraction of labeled data and a large amount of unlabeled data (Jeong et al. (2020); Zhao et al. (2022)). ③ Labels-at-Partial-Clients, where a few clients are fully labeled while others are unlabeled (Li et al. (2023); Liang et al. (2022); Liu et al. (2021; 2024); Zhang et al. (2024)). Our work focuses on the Label-at-All-Client setting. Recent research (Cho et al. (2023); Zhang et al. (2023); Bai et al. (2024); Zhu et al. (2024); Liu et al. (2025)) builds on FixMatch (Sohn et al. (2020)), focusing on pseudo-label selection, debiasing or combination. However, with both internal and external heterogeneity, these methods cannot avoid fewer data participation since they discard low-confidence samples as FixMatch and are also hard to fit the real global distribution by the indirect statistics.

## C THEORETICAL PROOFS

In this section, we provide the convergence analysis for the bi-level optimizations of ProxyFL: Global Proxy Tuning (GPT) and Indecisive-Categories Proxy Learning (ICPL). Our proofs are based on the standard assumptions in the non-convex optimization.

### C.1 CONVERGENCE OF GLOBAL PROXY TUNING

Our GPT module is a global optimization process executed on the server. In each communication round, the server collects the local proxies $\{\omega_m\}_{m=1}^M$ from clients and then optimizes the global proxies $\boldsymbol{\Omega}_{\mathcal{G}}$ by minimizing the loss function $\mathcal{L}_{GPT}$. First, we give:

**Theorem 1 (Convergence of Global Proxy Tuning)** *Assume that the loss function $\mathcal{L}_{GPT}$ is L-Lipschitz and bounded below, where $\mathcal{L}_{GPT}$ is related to $\boldsymbol{\Omega}_{\mathcal{G}}$. By optimizing the global proxies $\boldsymbol{\Omega}_{\mathcal{G}}$ via gradient descent with learning rate $\eta_{\mathcal{G}}$ such that $0 < \eta_{\mathcal{G}} \leq \frac{1}{L_{\mathcal{G}}}$, the optimization process converges to a stationary point. I.e.,*

$$\lim_{Q \to \infty} \frac{1}{Q} \sum_{q=0}^{Q-1} \mathbb{E}\left[\left\|\nabla_{\boldsymbol{\Omega}_{\mathcal{G}}} \mathcal{L}_{GPT}\left(\boldsymbol{\Omega}_{\mathcal{G}}^q\right)\right\|^2\right] = 0, \tag{13}$$

*where $Q$ is the number of optimization steps on the server.*

Then we provide a specific proof for Theorem 1. According to the descent lemma for L-smooth functions, we have:

$$\mathcal{L}_{GPT}(\boldsymbol{\Omega}_{\mathcal{G}}^{q+1}) \leq \mathcal{L}_{GPT}(\boldsymbol{\Omega}_{\mathcal{G}}^{q}) + \langle \nabla_{\boldsymbol{\Omega}_{\mathcal{G}}} \mathcal{L}_{GPT}(\boldsymbol{\Omega}_{\mathcal{G}}^{q}), \boldsymbol{\Omega}_{\mathcal{G}}^{q+1} - \boldsymbol{\Omega}_{\mathcal{G}}^{q} \rangle + \frac{L_{\mathcal{G}}}{2} ||\boldsymbol{\Omega}_{\mathcal{G}}^{q+1} - \boldsymbol{\Omega}_{\mathcal{G}}^{q}||^2. \tag{14}$$

According to the gradient-descent formula $\boldsymbol{\Omega}_{\mathcal{G}}^{q+1} = \boldsymbol{\Omega}_{\mathcal{G}}^{q} - \eta_{\mathcal{G}} \nabla_{\boldsymbol{\Omega}_{\mathcal{G}}} \mathcal{L}_{GPT}(\boldsymbol{\Omega}_{\mathcal{G}}^{q})$, Eq. 14 can be re-written as:

$$\mathcal{L}_{GPT}(\boldsymbol{\Omega}_{\mathcal{G}}^{q+1}) \leq \underbrace{\mathcal{L}_{GPT}(\boldsymbol{\Omega}_{\mathcal{G}}^{q}) - \eta_{\mathcal{G}} ||\nabla_{\boldsymbol{\Omega}_{\mathcal{G}}} \mathcal{L}_{GPT}(\boldsymbol{\Omega}_{\mathcal{G}}^{q})||^2 + \frac{L_{\mathcal{G}} \eta_{\mathcal{G}}^2}{2} ||\nabla_{\boldsymbol{\Omega}_{\mathcal{G}}} \mathcal{L}_{GPT}(\boldsymbol{\Omega}_{\mathcal{G}}^{q})||^2}_{} \tag{15}$$

$$= \mathcal{L}_{GPT}(\boldsymbol{\Omega}_{g}^{q}) - \eta_{\mathcal{G}} (1 - \frac{L_{\mathcal{G}} \eta_{\mathcal{G}}}{2}) ||\nabla_{\boldsymbol{\Omega}_{\mathcal{G}}} \mathcal{L}_{GPT}(\boldsymbol{\Omega}_{\mathcal{G}}^{q})||^2$$

Let the learning rate $\eta_{\mathcal{G}} \leq \frac{1}{L_{\mathcal{G}}}$, such that $1 - \frac{L_{\mathcal{G}} \eta_{\mathcal{G}}}{2} \geq \frac{1}{2}$. Thus, Eq. 15 can be simplified to:

$$\mathcal{L}_{GPT}(\boldsymbol{\Omega}_{\mathcal{G}}^{q+1}) \leq \mathcal{L}_{GPT}(\boldsymbol{\Omega}_{\mathcal{G}}^{q}) - \frac{\eta_{\mathcal{G}}}{2} ||\nabla_{\boldsymbol{\Omega}_{\mathcal{G}}} \mathcal{L}_{GPT}(\boldsymbol{\Omega}_{\mathcal{G}}^{q})||^2 \tag{16}$$

Rearranging the terms, we get:

$$||\nabla_{\boldsymbol{\Omega}_{\mathcal{G}}} \mathcal{L}_{GPT}(\boldsymbol{\Omega}_{\mathcal{G}}^{q})||^2 \leq \frac{2}{\eta_{\mathcal{G}}} (\mathcal{L}_{GPT}(\boldsymbol{\Omega}_{\mathcal{G}}^{q}) - \mathcal{L}_{GPT}(\boldsymbol{\Omega}_{\mathcal{G}}^{q+1})) \tag{17}$$

Summing both the left and right sides of Eq. 17 from $q = 0$ to $Q - 1$, we have:

$$\sum_{q=0}^{Q-1} ||\nabla_{\boldsymbol{\Omega}_{\mathcal{G}}} \mathcal{L}_{GPT}(\boldsymbol{\Omega}_{\mathcal{G}}^{q})||^2 \leq \underbrace{\frac{2}{\eta_{\mathcal{G}}} \sum_{q=0}^{Q-1} (\mathcal{L}_{GPT}(\boldsymbol{\Omega}_{\mathcal{G}}^{q}) - \mathcal{L}_{GPT}(\boldsymbol{\Omega}_{\mathcal{G}}^{q+1}))}_{} \tag{18}$$

$$= \frac{2}{\eta_{\mathcal{G}}} (\mathcal{L}_{GPT}(\boldsymbol{\Omega}_{\mathcal{G}}^{0}) - \mathcal{L}_{GPT}(\boldsymbol{\Omega}_{\mathcal{G}}^{Q}))$$

Since we adopt InfoNCE loss (Oord et al. (2018)) for $\mathcal{L}_{GPT}$ (See Sec. 3.2.1) with its lowerbound 0, we thus have:

$$\sum_{q=0}^{Q-1} ||\nabla_{\boldsymbol{\Omega}_{\mathcal{G}}} \mathcal{L}_{GPT}(\boldsymbol{\Omega}_{\mathcal{G}}^{q})||^2 \leq \frac{2}{\eta_{\mathcal{G}}} \mathcal{L}_{GPT}(\boldsymbol{\Omega}_{\mathcal{G}}^{0}) \tag{19}$$

Dividing both sides by $Q$ and taking the limit, we have:

$$\lim_{Q \to \infty} \frac{1}{Q} \sum_{q=0}^{Q-1} ||\nabla_{\boldsymbol{\Omega}_{\mathcal{G}}} \mathcal{L}_{GPT}(\boldsymbol{\Omega}_{\mathcal{G}}^{q})||^2 \leq \lim_{Q \to \infty} \frac{2 \mathcal{L}_{GPT}(\boldsymbol{\Omega}_{\mathcal{G}}^{0})}{\eta_{\mathcal{G}} Q} = 0 \tag{20}$$

Since the squared norm of the gradient (*i.e., the left-hand side of Eq. 20*) is non-negative, **hence we have proven Theorem 1.**

## C.2 CONVERGENCE OF LOCAL TRAINING WITH ICPL

The ICPL module is executed during the local training on each client. The total local loss is denoted as $\mathcal{L}_{local} = \mathcal{L}_s + \alpha \mathcal{L}_u + \beta \mathcal{L}_{ICPL}$, with optimization parameters $\boldsymbol{\Theta}_k = \theta_k \cup \omega_k$.

**Theorem 2 (Convergence of Local Training with ICPL)** *Suppose that the total local loss function $\mathcal{L}_k$ for client $k$ is L-smooth and bounded below, where $\mathcal{L}_k$ is related to $\boldsymbol{\Theta}_k$. By optimizing the local model parameters $\boldsymbol{\Theta}_k$ via gradient descent with learning rate $\eta_l$ such that $0 < \eta_l \leq \frac{1}{L_k}$, the optimization process converges to a stationary point. I.e.,*

$$\lim_{E \to \infty} \frac{1}{E} \sum_{e=0}^{E-1} \mathbb{E}[||\nabla_{\boldsymbol{\Theta}_k} \mathcal{L}_k(\boldsymbol{\Theta}_k^e)||^2] = 0 \tag{21}$$

*where $E$ is the number of local training epochs.*

Similar to Sec. C.1, following the descent lemma and the gradient-descent update formula, we can similarly derive:

$$\mathcal{L}_k(\mathbf{\Theta}_k^{e+1}) \leq \mathcal{L}_k(\mathbf{\Theta}_k^e) - \eta_l(1 - \frac{L_k\eta_l}{2})\|\nabla_{\mathbf{\Theta}_k}\mathcal{L}_k(\mathbf{\Theta}_k^e)\|^2 \tag{22}$$

By setting a local learning rate $\eta_l \leq \frac{1}{L_k}$, we obtain:

$$\sum_{e=0}^{E-1} \|\nabla_{\mathbf{\Theta}_k}\mathcal{L}_k(\mathbf{\Theta}_k^e)\|^2 \leq \frac{2}{\eta_l}(\mathcal{L}_k(\mathbf{\Theta}_k^0) - \mathcal{L}_k(\mathbf{\Theta}_k^E)) \leq \frac{2}{\eta_l}\mathcal{L}_k(\mathbf{\Theta}_k^0) \tag{23}$$

Taking the limit to Eq. 23 and knowing that the squared norm is non-negative, we have:

$$0 \leq \lim_{E\to\infty} \frac{1}{E} \sum_{e=0}^{E-1} \|\nabla_{\mathbf{\Theta}_k}\mathcal{L}_k(\mathbf{\Theta}_k^e)\|^2 \leq \lim_{E\to\infty} \frac{2}{\eta_l E}\mathcal{L}_k(\mathbf{\Theta}_k^0) = 0 \tag{24}$$

Thus,

$$\lim_{E\to\infty} \frac{1}{E} \sum_{e=0}^{E-1} \|\nabla_{\mathbf{\Theta}_k}\mathcal{L}_k(\mathbf{\Theta}_k^e)\|^2 = 0 \tag{25}$$

**So we have proven Theorem 2.**

## D LLM USAGE CLAIM

We only utilize Large Language Models (LLMs) to polish a few sentences in this manuscript. More importantly, all these sentences have been subsequently revised manually by us.

