# OpenReview forum: "ProxyFL: A Proxy-Guided Framework for Federated Semi-Supervised Learning"
_ICLR.cc/2026/Conference — ICLR 2026 Conference Withdrawn Submission_

### Official Review · Reviewer_58A1 · 2025-10-26

**Soundness:** 3
**Presentation:** 3
**Contribution:** 2
**Rating:** 6
**Confidence:** 4

**Summary:**

The authors proposed two techniques for Federated semi-supervised learning to mitigate the  heterogeneity issue, namely a Global Proxy-Tuning (GPT) and IndecisiveCategories Proxy Learning (ICPL) .  The presentation is clearly. Experimental results and ablation study are solid.

**Strengths:**

1. The authors proposed two techniques for Federated semi-supervised learning to mitigate heterogeneity, namely a Global Proxy-Tuning (GPT) and IndecisiveCategories Proxy Learning (ICPL) . The topic is relevant and important
2. The presentation is clear. The author generally have a clean problem formulation and math notation.
3. Clean thought flow. The author starts from observations from experiments, then raised questions and propose solution.
4. Experimental results and abliation study are solid.

**Weaknesses:**

1. I respectfully believe, the contribution of the methods is slightly low. For Global Proxy-Tuning (GPT), the idea -- averaging local weights with the considerations of outliers--, have been expored by multiple existing works for federated supervised learning. The Indecisive Categories Proxy Learning (ICPL) is essentially a coarse labeling process, commonly adopted in semi-supervised learning.

2. I would ecourage author briefly analyze the impact of two methods on global learning dynamic, assuming all (x, y) follow same distribution from P_{XY}. I understand the challenge, i.e., label changing in semi-supervised learning.


3. The name ProxyFL is a little bit confusing. The author does not apply any proxy gradient method in the paper.

**Questions:**

1. Why not use the GPT for the whole model but only for the final classifier?
2. Is there anything special in the convergence analysis in the appendices.
3. Would appreciate if the authors could conduct experiment on centrialized semi-supervised learning approach ?
4. Would appreciate if the authors could try some other models, like resnet20, 50 or transformer

---

### Official Review · Reviewer_ngfs · 2025-10-31

**Soundness:** 3
**Presentation:** 3
**Contribution:** 2
**Rating:** 4
**Confidence:** 4

**Summary:**

The paper proposes ProxyFL, a framework for addressing both external and internal heterogeneity in federated semi-supervised learning using a unified proxy mechanism. ProxyFL treats classifier weights as category proxies and optimizes global proxies to align with clients’ label distributions, thereby mitigating external heterogeneity. To handle internal heterogeneity from unreliable pseudo-labels, it introduces indecisive-category proxy learning, which better utilizes low-confidence unlabeled samples.

**Strengths:**

- Addresses both external and internal heterogeneity with a unified proxy.
- Paper is well written and easy to follow.

**Weaknesses:**

- Requiring knowledge of global labeled-data distribution may still raise privacy concerns, particularly for rare class distributions or small client populations.
- Efficiency evaluation is missing. ProxyFL introduces additional server-side computation and utilizing more unlabeled data, which results in more computation. Communication overhead and convergence analysis with wall-clock should be compared against baselines, not only epoch-level metrics.
- Lacks comparison with existing method. For example, Twinsight [w1] tackles similar problem but not compared as baseline.
- More ablation study and sensitivity analysis would be beneficial. I will detail in questions section.

[w1] Zhang, Yonggang, et al. "Robust Training of Federated Models with Extremely Label Deficiency." The Twelfth International Conference on Learning Representations.

**Questions:**

- How large additional server-side overhead does ProxyFL introduces? Is it scalable to larger client pools?
- Number of local epochs and server-side epochs would have a huge impact on training behavior. It would be better to have an ablation study on that.
- How did $\alpha$ and $\beta$ values are decided? Any rationale behind that?
- How does ProxyFL perform with different ratios of labeled data other than 10%?
- Some empirical insights are expected/trivial (Fig. 2-c, more ground-truth labels -> higher accuracy). More informative scenarios include keeping the same labeled ratio while increasing unlabeled data with controlled pseudo-label quality.
- In Sec. 3.2.2 (line 259), the claim that incorrect pseudo-labels still avoid irrelevant categories is not sufficiently supported. Could you provide additional analysis or references?
- For Fig. 2-b, could the authors clarify at which epoch the accuracy difference is measured and consider reporting percentages for clarity?


Minor
- The description of Fig. 1(a)-(b) does not fully align with the text (line 73).
- Fig. 2-b: The y-axis would be clearer in percent; also specifying which epoch is plotted would be informative.
- Line 280: "It" should be "its."

---

### Official Review · Reviewer_Bipd · 2025-10-31

**Soundness:** 3
**Presentation:** 2
**Contribution:** 3
**Rating:** 4
**Confidence:** 4

**Summary:**

This paper presents a novel approach for Federated Semi-Supervised Learning by introducing ProxyFL, a proxy-guided framework that simultaneously mitigates both external and internal heterogeneity. The key innovation is using learnable classifier weights as proxies to model the category distribution both locally and globally. The method addresses the challenges of data heterogeneity, providing a new solution to the limitations of existing FSSL methods. The experiments demonstrate the effectiveness of ProxyFL, showing improvements in performance and convergence speed over several baseline methods.

**Strengths:**

1.Originality: The use of proxies to address both external and internal heterogeneity in FSSL is a highly original and valuable contribution. The idea of using learnable classifier weights as proxies to model category distributions both locally and globally is innovative and offers a scalable solution.
2.Quality: The experiments are thorough and demonstrate significant improvements over existing methods. The paper presents a clear explanation of the ProxyFL framework, backed by extensive empirical evidence.
3.Clarity: The writing is generally clear and accessible, with a logical flow. The experiments and results are well explained, though some aspects, such as the dynamic proxy pooling, could be made clearer with additional figures.
4.Significance: The contribution addresses a critical problem in federated learning—data heterogeneity, and provides a solution that could have broad applications in decentralized machine learning systems.

**Weaknesses:**

1. While the proposed ProxyFL framework shows great potential, the theoretical discussion could be further expanded. Specifically, a more detailed explanation of why proxies are effective in mitigating both external and internal heterogeneity would strengthen the paper. It would be helpful to explore how proxies help the model adapt to data distributions under varying levels of heterogeneity.
2.While the paper introduces key mechanisms like the Dynamic Proxy Pool and Global Proxy Tuning , these complex processes are not sufficiently visualized. Adding more detailed diagrams, such as how the dynamic proxy pool is constructed or how GPT optimizes global proxies, would help readers better understand these core components.
3.Although the experiments are well-conducted, since the field typically uses these datasets, further experiments with hyperparameter tuning on these datasets could be valuable.

**Questions:**

1.The paper shows that the proxy approach effectively mitigates both external and internal heterogeneity, but could the authors provide a more detailed explanation of the theoretical motivation behind the proxy design? Specifically, why does this method work well across different data heterogeneity conditions?
2.Can the authors provide more discussion about hyperparameter sensitivity, especially with respect to learning rate, confidence threshold, and other settings?
3. This paper has conducted experiments on several datasets under different heterogeneity conditions, but could the authors conduct further tuning experiments to explore the robustness of the method across a range of hyperparameter settings, particularly under extreme conditions?

---

### Official Review · Reviewer_L3BF · 2025-11-01

**Soundness:** 3
**Presentation:** 2
**Contribution:** 3
**Rating:** 4
**Confidence:** 4

**Summary:**

The paper introduces ProxyFL, a federated semi‑supervised learning (FSSL) framework that addresses both external heterogeneity (across clients) and internal heterogeneity (within each client). The method treats the classifier’s learnable weights as proxies for class distributions at both local and global levels. On the server, a pull–push–style global proxy optimization aims to align the global proxy with the true cross‑client category distribution. On clients, an indecisive‑category proxy learning mechanism incorporates low‑confidence unlabeled samples using a positive–negative proxy pool, with the goal of reducing errors from unreliable pseudo‑labels. Experiments suggest improved performance and convergence over existing FSSL baselines.

**Strengths:**

1. Unified treatment of heterogeneity. The proxy formulation provides a single mechanism to address both cross‑client (external) and within‑client (internal) distributional challenges.

2. Role separation between server and clients. The server focuses on global proxy tuning, while clients use indecisive‑category proxy learning—a clear division of responsibilities.

3. Empirical gains. The method demonstrates competitive or superior results in the presented experiments, with indications of faster convergence.

**Weaknesses:**

1. **Positioning of global proxy tuning.** The paper should more clearly articulate how the proposed global proxy differs from prior server‑side prototype/centroid refinement—e.g., Orchestra’s globally consistent clustering and centroid updates [1].
- What is unique about optimizing classifier weight vectors as proxies (objective, constraints, and update rules) versus updating global centroids/prototypes?
- Are there theoretical or empirical reasons why class‑weight proxies better capture global category structure than prototype aggregation?

2. **Indecisive‑category selection and Eq. (4) (global‑prior thresholding).** The use of a global label distribution to set logit thresholds requires deeper justification and analysis.
- Effectiveness vs. alternatives. Is this significantly better than a uniform threshold? Please add a direct ablation.
- Privacy considerations. Aggregating per‑class labeled counts across clients can leak information.
- Label vs. unlabeled distribution mismatch. A global prior based only on labeled data can be biased when unlabeled data follow a different distribution.
- Few‑label robustness. When labeled data are very limited (as in (FL)2 [2]), how stable are the thresholds? Please include sensitivity curves vs. the number of labeled samples per class.
- Why is it called a dynamic threshold? Does it mean that thresholds change over time?

3. **Experimental protocol and cross‑device realism.** The current setup uses 20 clients with 40% participation per round. In cross‑device FSSL, it is common to have many more clients and low participation rates (~5–10%).
- Please evaluate ProxyFL under larger‑scale, low‑participation regimes and report stability of global proxy tuning under pronounced per‑round heterogeneity.

[1] E. Lubana et al., "Orchestra: Unsupervised Federated Learning via Globally Consistent Clustering," ICML, 2022.
[2] S. Lee et al., "(FL)2: Overcoming Few Labels in Federated Semi-Supervised Learning," NeurIPS, 2024.

**Questions:**

Please see the weaknesses

---

### Official Review · Reviewer_hrRt · 2025-11-02

**Soundness:** 2
**Presentation:** 2
**Contribution:** 3
**Rating:** 4
**Confidence:** 3

**Summary:**

This paper focuses on the core challenge of Federated Semi-Supervised Learning (FSSL) —— data heterogeneity(including external heterogeneity across clients and internal heterogeneity within clients) and proposes a proxy-guided framework called ProxyFL. The core design of Global Proxy-Tuning (GPT) and Indecisive-Categories Proxy Learning (ICPL) is proposed to mitigate both types of heterogeneity simultaneously.
Core Contributions:
This paper is the first to propose a "unified proxy" mechanism that mitigates both external and internal heterogeneity in FSSL. It verifies the effectiveness and convergence of the method under different heterogeneity levels through empirical and theoretical analyses.

**Strengths:**

1.The paper has a strong focus on addressing key pain points: it centers on the core bottlenecks of external heterogeneity and internal heterogeneity in FSSL and proposes corresponding solutions. The comparison methods encompass three categories: FL, FL+SSL and FSSL, ensuring comparability.
2.This paper introduces the concept of “classifier weights as a unified proxy”, simultaneously addressing both internal and external heterogeneity. The GPT optimization logic is reasonably sound, the ICPL design mechanism is flexible, and it incorporates a theoretical proof of convergence, providing solid theoretical support.

**Weaknesses:**

1.The core differences between ProxyFL and PCL have not been sufficiently elaborated. It is necessary to explain the scenario adaptation innovations achieved by applying PCL to FSSL, rather than merely a direct combination of methods.
2.The advantage of "unification" has not been verified.The performance difference between the "unified proxy” and the hybrid schemes of “GPT + existing internal heterogeneity processing method” or “existing external heterogeneity processing method + ICPL” has not been compared.
3.Technical details are not fully explained. The construction logic for positive and negative proxy pools in ICPL lacks an intuitive explanation (Equations 5 and 7).
4.Writing details and chart details require refinement.
5.The choice of loss function lacks justification, with no explanation provided for why GPT employs the InfoNCE loss over other loss functions.
6.The privacy protection logic contains certain vulnerabilities.

**Questions:**

1.It is recommended to strengthen the contrast with PCL and highlight the innovative aspects of ProxyFL in the FSSL scenario.
2.Ablation experiments comparing “Unified Proxy vs. Hybrid Methods” are required to validate the “1+1 > 2” effect of ProxyFL (GPT+ICPL).
3.A flow chart should be supplemented in Fig. 3 to visually demonstrate the construction process of the positive-negative proxy pool,reducing the comprehension burden.
4.Terminology requires immediate annotation. For example, the terms “external heterogeneity” and “internal heterogeneity” appear for the first time in the abstract and main text. The details of the figures require further refinement. For example, the outliers in Fig. 2(a) should be clearly labeled.
5.It is recommended to supplement ablation experiments on the loss function to demonstrate GPT's computational efficiency and robustness when employing InfoNCE.
6.The paper claims that “proxy does not compromise privacy,” but the global prior of ICPL requires collecting the number of category labels from all clients. Therefore, it is necessary to clarify whether the transmission of this statistical information complies with privacy protection.

---

### Note · Authors · 2025-11-14

I have read and agree with the venue's withdrawal policy on behalf of myself and my co-authors.